# Leadership Theories and the Veterinary Health Care System

**DOI:** 10.3390/vetsci9100538

**Published:** 2022-09-29

**Authors:** Holger Fischer, Petra Heidler, Lisa Coco, Valeria Albanese

**Affiliations:** 1Tierärztliches Kompetenzzentrum für Pferde Großwallstadt Altano GmbH, 63868 Großwallstadt, Germany; 2Faculty of Health and Medicine, Department for Economy and Health, University for Continuing Education Krems, 3500 Krems, Austria; 3Department of Health Sciences, St. Pölten University of Applied Sciences, 3100 St. Pölten, Austria; 4Department of International Business and Export Management, IMC University of Applied Sciences Krems, 3500 Krems, Austria

**Keywords:** leadership theory, leadership, transformational leadership, veterinary

## Abstract

**Simple Summary:**

Good leadership is necessary for the success of every company. The health care industry, including the veterinary sector, has been increasingly acknowledging the importance of an appropriate leadership, and the attention towards leadership theories in this field has steadily increased in the literature through the past one and a half decade. However, a consensus on what effective leadership is and on how it can contribute to the veterinary health sector is still lacking. This review aims to describe the most prominent leadership theories as well as leadership styles, as they may apply to the veterinary health care sector and to discuss transformational leadership in routine and crisis situation in the human and veterinary health care industry.

**Abstract:**

This descriptive review aims to illustrate the different leadership theories as they may apply to the veterinary health care sector, and specifically, to the affection and sports animal subsectors. The increasing and ever-changing challenges veterinary health care operators and investors operating in these subsectors face are briefly described, as well as the most known leadership theories and styles, each with its possible advantages and disadvantages specific to its application to the veterinary health care system. The different theories are illustrated in their key aspects and their historical evolution. Finally, the discussion focuses on transformational leadership as it is seen as the most progressive and promising leadership style to hold up to said challenges in the veterinary health care system.

## 1. Introduction

Regardless of their business area and regardless of their size, companies benefit of good leaders to express their full potential and to thrive.

In particular, the health care sector is dynamic and complex. It experiences its own peculiar problems and challenges as well as those of organizations in other sectors [1,2,3]. Examples are the increasing expectations for transparency, the increasing threats of liability, the increasing influence of political groups and social media, and the shortage of young professionals willing and capable of replacing their aging counterparts, just to name a few [4,5,6]. External factors, such as government policies, globalization, changing population demographics, economic factors, progressing, and advancing medical technologies also have a substantial and increasing influence on the health care system [7,8,9]. The health care sector in these historical times is experiencing change at a pace that was not seen before. This change is a product of both internal and external factors that are largely outside the area of expertise of most health professionals. The health care industry is, therefore, calling for changes and reforms to meet the increasing expectations of patients and clients, the most important of which are safety, transparency, accountability, efficiency, and quality [10]. Cutting edge leadership is essential in establishing a vision for health care businesses and organizations, and in implementing necessary reforms and changes. Indeed, the topics “leadership” and “management” have received an unprecedented attention in the medical field in multiple countries since the turn of the century [3,11,12,13,14,15,16,17,18,19,20,21,22,23,24].

The primary goal of this review is to raise awareness on the topic of leadership in veterinary medicine.

Secondly, the authors aimed to review the major leadership theories and discuss, based on their experience, how they could apply to the veterinary health care sector.

Third and most importantly, the authors try to advise the readers on how the establishment of a conscious leadership culture could be helpful in facing the present and upcoming challenges of the veterinary profession, in daily practice management as well as in crisis management.

A very important aspect that is being investigated in the field of health care leadership is that of the skills and characteristics that the ideal leader should possess to be successful and to make their company thrive. Identifying these traits could help in the recruiting of potential leaders and also, perhaps more importantly, in the education and training of leaders-to-be, aimed at improving the team and organization performance in the medium to long run [22].

## 2. Literature Review

What is a leader and what does a leader do? What is expected from a successful leader in the health care sector, and especially in the veterinary health care system?

The Oxford dictionary defines a leader first as *A person in control of a group, a country or a situation.* This definition is tautological and not exhaustive, it does not explain in which sense a leader “ontrols”, what tools they use to exert their control and why are they in such a position.

The same source continues: *A person who manages or controls other people, especially because of his or her ability or position.* This definition is more complete and starts to shine a light on what we will see as one of the most interesting aspects of this field of research: their *ability*. So, it appears that a leader is not only someone who sits on the leader chair, but someone who, in order to sit on said chair, needs to have something that others do not have [25].

Several leadership theories have been developed and researched through the years. Some of them have fallen out of favor, some new ones are developed continuously. A leadership theory is a slightly different concept than that of a leadership style.

“Theory provides the overarching sense-making frame for experience. Without a theoretical framework to connect and integrate experiences there is no sense-making, and thus there can be no learning” [26]. Leadership theories answer the questions of why and how some people become leaders and some do not.

Leadership theories tend to focus either on the traits or on the actions and demeanor that people should or should not display in order to be good leaders.

Over the years, the subject has been looked at from many different perspectives, and four prominent groups of theories can be identified. Newer leadership theories do not replace or discard the old ones, but rather they enrich and further on elaborate them, and look at them in a different way. Therefore, much of the information provided in this classification might appear redundant.

There are dozens of leadership theories, the most famous of which can be classified in four main groups: trait theories, functional theories, behavioral and style theories, and contemporary theories (Table 1).

### 2.1. Trait Theory

The trait theory, also known as the Great Man Theory, sustains that good leaders are born, not made. These people are destined and deserve to be good leaders thanks to their inborn special characteristics, special talents, and special capabilities, that cannot be taught or learned. Kohs and Irle [27], Bernard [28], Bingham [29], Tead [30], and Kilbourne [31] all explained leadership in definition of traits of character and personality [32]. By the 1950s, hundreds of studies had already been conducted exploring these traits [33]. Later on, these theories evolved to contemplate the possibility that such characteristics can be developed in others who were showing an innate disposition to them, and if not exactly taught or learned, at a minimum, they could be enhanced and further implemented.

Based on several research studies, Bass [32] developed a profile of traits that are evident in successful leaders. These are categorized into three areas:Intelligence, including judgement, decisiveness, knowledge, and fluency.Personality, including adaptability, alertness, integrity, and nonconformity.Ability, including cooperativeness, popularity, and tact.

A spin-off of the Great Man Theory identifies the leader based on their charisma. Suasiveness, convincingness, personal appeal, seductiveness, aplomb, and whit are the main features of charismatic leaders. The leader’s personality leverages loyalty and commitment, first to the person of the leader themselves, and second to the aims and goals of the organization that they represent. It is important to note that charismatic leadership is attributed based on what people *perceive and interpret* of the leader’s behavior, and not by an objective set of characteristics and traits. Furthermore, one “charismatic” trait is not sufficient to identify a leader as charismatic, rather, there must be a constellation of traits and/or behaviors in order to entitle them to this attribute. Charisma, therefore, remains a concept that may seem easy to grasp but is very difficult to precisely define [34].

### 2.2. Functional Theory

Subsequent to trait theories of leadership, the functional theory of leadership was developed. This theory underscores the functions of the leader (and not their traits) and bypasses the dilemma between appointed vs. naturally emerging leader. It indicates that, in either case, the leader can and should learn to elicit the right functions at the right time. Kotter [35] suggests that, by focusing on the functions of the leader, their performance can be improved by training and, thereby, their leadership skills can further develop and be perfected. The same author also indicates that organizations should not wait for leaders to come along, but instead they should grow and train their own, by identifying employees with leadership potential and enabling them to develop those skills. This approach, therefore, also supports the idea that several members of a professional group can strive to develop their leadership potential. As an example, one widely accepted functional theory is Adair’s action-centered leadership, according to which the leader has to take into account and consider three sets of needs of the organization: the technical needs or the tasks that need to be conducted, the needs of each single team member or individual needs, and the team needs or what needs to be performed to keep the group together efficiently. How successful the leader is, is decided by how effectively and harmonically they can take care of these three areas. Too much attention or effort in one over the others can cause imbalance and can interfere with the productiveness and effectiveness of the group. This could consequently affect the spirit of the group, and therefore, the quality of its performance and the overall outcome and achievements [36].

### 2.3. Behavioral Theory

The third group of leadership theories focuses on several possible styles of leadership. A leadership style is the way in which the leader carries out their functions, that is, the way in which the leader is perceived as typically behaving towards their followers (hence the definition of behavioral theory) [37]. Such leaders develop their styles through education, training, and life experiences. Therefore „style” refers to how, as leaders, we behave on a day-to-day basis, because we know or have learnt from previous experience that the way we do things works. The behavioral leadership theory is based on the assumption that these operating standards are repeatable and other leaders can duplicate them. Behind this theory is the concept that no inborn trait makes a leader successful, but rather a set of behaviors that can be replicated, taught, and learnt. The core point of leadership according to this theory is how the leader acts and not what their qualities or talents per se are. Actions are, therefore, the best predictors for success. Patterns of behavior are investigated and grouped into categories defined as “leadership styles”.

In the perspective of behavioral theory, the potential leader gets to prospectively and somehow freely determine and plan their leadership style and act accordingly, which can be attractive. Nevertheless, leadership styles are fluid and flexible and can be adapted by changing patterns of behavior according to the circumstances. Furthermore, according to this theory, a leader does not necessarily possess innate traits that make them special, instead they become leader based on their actions, which in turn means that virtually anybody can become a leader, provided they behave as such. Of course, there is not a handbook that suggests a specific leadership style for every possible circumstance and every possible challenging situation in each and all possible industry, and it should not be expected that there is one. There are dozens of leadership styles that stem from the behavioral theory, but there is not a right one for every circumstance. The traditional classification of leadership styles within the behavioral theory is the following: authoritarian, democratic, permissive, and bureaucratic.

An authoritarian leader focuses on power and authority, exercises control and directive behavior, makes decisions alone and expects obedience of instructions, and uses coercion.

A democratic leader seeks, at least formally, the views of all relevant parties. They consult and work together with individuals and teams, engage in two-way communication, and encourage collaborative teamwork.

A permissive leader uses few established rules and policies. They monitor performance from a distance, which may make them seem detached, and permits individuals and teams to work autonomously.

A bureaucratic leader follows established policies and rules to the letter. Their rules are fixed and inflexible, and their communication is impersonal. Despite the approving and disapproving overtones of each of these styles of leadership, there are merits and weaknesses to each, depending on the situation, and especially in the health care setting.

Another way of classifying leaders based on the behavioral theory is task-oriented vs. people-oriented leader. The first type of leader will address problems always starting from within the workflow. The second type of leader, instead, will address problems starting from the individuals first. A leader who is too focused on people may be popular but not so productive, whereas a leader who is too focused on tasks may get a lot of work done, however the employees may be unhappy [37].

### 2.4. Contingency Theory

The contingency leadership theories, sometimes called situational theories, focus on the context of a leader. In the perspective of situational leadership theories, the efficacy of a leader has little to do with their personality, but it is rather a product of the circumstances the leader operates under. According to situational theories, changes in leadership styles dictated by the context are necessary for an effective leadership. In the interest of the further development of a company, these theories advise that there is a right leader for most specific circumstances, and that person should take over when the previous leader is no longer adapting to the present needs. Examples of contingency theories are the Hershey and Blanchard’s Situational Theory [38], the Evans and House Path-Goal Theory [39], and Fiedler’s Contingency Theory [40,41].

The contingency theories can be very inspiring. A very positive side to their perspective is that there would be an effective leadership for virtually any—no matter how challenging—context and situation. However, as it is often the case, there is not any very detailed source or handbook that one can consult to learn more details of specific situations and the challenges that it may present. Furthermore, contingency theories tend to focus on the context and the situation and to underscore them as they may be presented from factors outside the company itself, and in turn they may not give enough importance to the psychology of the employees and their wellbeing or to other types of inside factors. Moreover, these theories can appear somehow shallow as they fail to recognize that leadership evolves, also, because the person of the leader self-changes and grows and evolves in their role, independently from the circumstances. A leader is indeed influenced by a number of internal and external factors such as the type of company and what industry it is part of, how big the team is, how cohesive the team is, what type of person the leader is, and how they take success and failure are examples of internal factors that influence leadership. Economy, political events, social media influence, and customer reviews are examples of external factors that impact leadership [42].

### 2.5. Contemporary Theories

#### 2.5.1. Management Theory

The transactional leadership theory or management theory is very common in the educational system, in the corporate world and to some extent, also, in the health care system. Its focus lies ultimately on the final performance of the group, and how to achieve success through organization and effective supervision. Transactional leadership consists of a system of benefits and penalties; when employees do right, they get rewarded. When they fail, they get warned and eventually punished. This theory is based on the principle that employees can only be motivated by rewards, and only dissuaded by penalties all directed to them personally. This theory discards the idea that people can behave a certain way because it is inherently right, or avoid doing something because it is inherently wrong [43].

Leading with punishments and rewards can be incredibly effective. Employees tend to appreciate and respond readily to positive reinforcement as they feel motivated and encouraged to succeed. However, strictly transactional leadership is seen critically because of how demoralizing punishments can feel to motivated employees who may feel treated unfairly and condescended upon. It can also be seen as a simplified and childish way of leading which does not live up to the standards and challenges of the modern world.

#### 2.5.2. Participative Theory

Participative leadership or democratic leadership is not as represented as transactional leadership in the corporate world, but it is very common in the public health care system. Employees led in a democratic fashion are directly involved in the decision-making process of the organization they are part of, and the role of the leader is that of enabling and facilitating the communication, filtering the various suggestions and advice. The ultimate decision is still taken by the leader, but after the members of the team have had the chance of expressing themselves on the topic [44].

Intuitively employees will feel much more motivated when directly involved in decision-making, which is a clear advantage of this theory. However, detractors of this theory suggest that it may be an unnecessary, inefficient, and weak way of leading. Furthermore, leaders who lead this way may lose the focus on what the company as such actually needs, because they are too distracted by considering what everybody thinks and wishes and wants. Lastly, considering everyone’s opinion is time consuming. 

Bill Gates is a famous example of democratic or participative leader: participative leaders find a way to ask the team how they would manage a given situation. Employees in these organizations are encouraged to openly share and discuss their opinions. These leaders’ strength is that they are capable of effectively take all the suggestions, process them, discuss them, and summarize them, ultimately considering them in their final decision. Employees, especially if educated, usually like this type of leaders; this is very important to consider, because democratically led organizations may, in the long run, attract more qualified and educated employees than organizations led in a less participative fashion. In the veterinary health care system, where most leaders and employees share the same education and have the same academic background, this theory is particularly appreciated, although it becomes less and less effective as the size of the company increases.

#### 2.5.3. Power Theory 

At the root of the power leadership theory is how the leader uses their power and their position to achieve their goals and those of the company they lead. An example of this is the French and Raven’s Five Forms of Power [45], which describes how the power stemming from their person and from their position influences the conduct of the leaders and what outcomes they have. 

Intuitively, in the short run, leaders with great power can be convincing and under this type of leadership things generally get conducted fast and reasonably well. So, it is easy to think this leadership theory describes one of the most efficient approaches to the topic. Once again though we cannot forget that employees are a dynamic factor, and leadership will influence the type of employees a company attracts. Educated employees tend to not like power leadership. The employees most companies aim to recruit (educated, accountable, and responsible) want a leader who does not enforce power over them, but works with them, encourages them, and helps them grow. A power leader instead makes employees feel dominated and controlled [46].

This leadership approach can be successful in very structured settings where hierarchy and promotion are underscored, such as in the military world, or in situations where the workforce is not educated at all and benefits of a stronger approach. In medical field organizations, employees in power led companies often see that their only chance to gain a place in the sun is to gain power of their own at all costs. This can have tremendously negative consequences such as low morale and inhospitable atmosphere [46]; especially when the health of the patients is at stake, the attention of the team may in the process get diverted from the mission of the organization.

#### 2.5.4. Connective Theory

The focus of the connective leader lies on their interpersonal relationships and connections. The connective leadership theory underscores the fostering role of the leader in establishing collaborations within the organization to help achieve the goals of the organization itself. These leaders can exquisitely mentor their employees, allotting large slots of their time to this mission. These leaders believe in making work enjoyable for their team and employees and they are committed to creating and maintaining a conducive, virtuous working environment. Many educated and self-motivated employees can thrive under this type of leadership.

Charns and Tewksbury [47], who are key advocates of connective leadership, explained that the leader, in order to wire connections and to promote integration of mentees, needs to take the following steps:-Scouting human resources to find collaborators.-Creating and communicating a shared vision.-Finding a role for their collaborators that enhances their value and adds value to the team.-Facilitating communication.-Building and maintaining social interactions and comfort.-Defining positions and tasks.-Documenting contributions and rewarding them properly.-Structuring contributions and contributors formally when appropriate.

When led this way, employees trust and confide in their leaders. This kind of leadership also inspires workers to follow the leader’s example and step up to a leadership role as well. Employees with growth prospects thrive under this leadership type and retention times tend to be long. Criticism to this leadership is mainly that being appropriately connected with similar other experts or individuals with similar interests through networking is already a recognized attribute of most effective leaders and it would not be enough per se to endorse a leading role. Other critics come from the thought that this type of leader would be so focused on interpersonal relationships that they would refrain from addressing problems with the employees in the fear of losing them or losing their appeal on them. However, it is generally agreed upon that connective leaders are, actually, quite effective overall.

#### 2.5.5. Transformational Theory

Transformational leadership refers to the “ability of leaders to influence others by transforming their behavior without necessarily being in a position of authority” [48]. A systematic review of transformational leadership conducted by Holly and Igwee [49] identifies it as comprising intellectual stimulation by incentivizing new ideas, considering all employees individually, motivating and inspiring them, stimulating their creativity, conveying optimism and meaning for the mission of the organization, guiding the team in a coherent direction, acting exemplarily, and fostering a sense of belonging and motivation. Kouzes and Posner [50] identify the key attributes of ideal transformational leaders:Leading by example.Inspiring a shared vision.Challenging the process.Enabling the others to act.Encouraging the heart.

Transformational leadership is, as in any other leadership model, not free from criticism. It may be seen as quite manipulative to have employees feel like they are part of something, that they belong to something bigger, although they, actually, rarely have a say on the major strategic or financial plans of the company, and they mostly do not own any stakes of it. Furthermore, transformational leaders are often criticized for focusing on small groups of people they consider worth their efforts, to further develop them. Lastly, the influence transformational leaders may exert on their employees can be detrimental if they are ill-minded or if their intentions are evil. In other words, if the leader has bad plans, followers might be deceived and be lured to go with whatever decisions the leader makes.

## 3. Discussion

The human medical profession has changed dramatically in the last century: medical care has changed, the education required to become medical doctors has changed, professional duties and offered services are different than they were, medical institutes are larger, more structured, and more organized, and the lifestyle of medical doctors has changed. In contrast to this, the veterinary profession has remained, until 10 years ago, relatively static: reading James Herriot’s books we realize that not so long ago solo practitioners had comparable lifestyles, comparable duties, and comparable working hours to what they used to have almost a hundred years ago. In over 50 years of combined working experience in small and large animal practice in five countries, the authors have observed that many veterinarians, although thankfully not all of them, are quite reluctant to change and to innovate. The statement “we have always done it this way” backs up a variety of procedures who lack evidence base, both medically and from a managerial standpoint. Focusing on the latter, which is the core topic of this review, the authors have observed that the expectations on young workforce in terms of working hours, minimum salary, professional duties, and military acceptance of directions are unique and are also often justified by the thought that “we have always done it this way”. Moreover, often veterinary practices lack an official emergency duty roster, an official distribution of the working load and a clear definition of the roles that the participants are called to fulfill, possibly generating confusion, frustration, and conflicts.

Nevertheless, clients are increasingly more informed, they are more educated, and they want the best for their animals, who are in part family members, in part valuable athletes, and in part an economical asset.

To accommodate the increasing expectations of the clientele, and to constantly improve the service provided to the animals, the clients, and the employees, the veterinary profession needs to accept innovation. Part of it is recognizing that a leadership culture is needed and must be established.

The abovementioned theories have advantages and disadvantages that make them more or less applicable to what the modern veterinary health care system needs to be.

The Great Man Theory has per se both proponents and critics. The main critic is that in its core, pure interpretation it fails to give credit to any learning process, work, or effort needed to become a leader. The Great Man Theory negates the modern concept that “you can if you want”. Moreover, the traits are difficult to define clearly, and it is questionable whether a single individual could possess all traits identified in the various studies. Furthermore, most personality features considered to be associated with good leadership skills are masculine, which is also seen critically, and do not really match the actual description of countless leaders who have been successful [51].

Proponents of this theory cite giants as Abraham Lincoln or Alexander the Great or Queen Elizabeth as their examples, as these people utilized their inherited skills, their *traits* indeed, to lead armies and countries. Furthermore, leaders are often very ambitious and determined, observation that appears to support this theory, nevertheless.

Kouzes and Posner [50] identified in a large-scale questionnaire the following characteristics associated with leaders perceived as effective: honest (89% of respondents), forward looking (71%), competent (69%), and inspiring (65%). A lesser number of respondents mentioned the following features: intelligent, broad-minded, fair-minded, dependable, supportive, straightforward, cooperative, and determined. An even lesser number mentioned: courageous, ambitious, caring, loyal, imaginative, mature, self-controlled, and independent. This variety of responses shows that there is no universal agreement on the nature of a leader or on the innate features that are necessary for a good leader, observation that challenges the trait theory. However, based on the authors’ experience, in the health care system, and more so in veterinary medicine, there are dozens of examples of solo practitioners, armed almost exclusively with strength, will, and grit, who founded a small clinic that progressively grew, acquired more employees, and became successful—a “great veterinarian” theory.

The risk associated with this theory, as it is applied to the veterinary world, is that of creating a practice that is centered around its leader and that it cannot function in their absence, which becomes a problem when the leader needs to take a leave for any reason.

The functional theory of leadership was innovative as it was developed in contrast to the trait theory, meaning that it underscored the importance of what the leaders do as opposed to what their innate characteristics are. This point has been largely absorbed and expanded by more modern theories to the point that the functional theory per se seems insufficient for the challenges presented by the contemporary veterinary health care system. In the authors’ opinion, nonetheless, two points of this theory remain valid: the necessity to identify employees with leadership potential, to grow them and to train them, and the leader’s obligation to juggle the three sets of need of the organization (technical, task, and individual).

The four major leadership styles, described by the behavioral theory, remain important in that any leader behaves in a way that falls predominantly in one of those categories: authoritarian, democratic, permissive, and bureaucratic, regardless of what their vision is, i.e., if they are transformational leaders or transactional leaders, for example.

The authors would like to provide the example of a leader of a local branch of a small animal corporate practice. Said leader may have an authoritarian style and like to personally assign every appointment to the staff veterinarian they consider most appropriate for each case. The central administration may also have transactional policies in place, such as a commission on top of the base salary, which the branch leader will apply.

Another example would be that of an equine hospital whose leader has a permissive style and a transformational vision; the staff veterinarian on call will take in an emergency, based on the duty roster. The leader will not intervene in any of the decisions or communications of the staff veterinarian unless requested. The performance and outcome of the staff veterinarian on call will be monitored from a distance, without direct interference. The leader will constantly reassure the employee on their ability of facing the situation and will offer advice and technical support at all times. Constructive, enthusiastic, and personalized feedback will be given by the leader in what the staff veterinarians have done properly. Criticism, where necessary, should not be avoided but it should never be aimed at humiliating or demeaning the employee, rather, at inspiring them to elevate their performance.

The contingency theory has been largely absorbed by subsequent theories. Nevertheless, it is very relevant to remind veterinary leaders how necessary changes in leadership styles can be when the context also changes. For example, the current paucity of young workforce in veterinary medicine has forced many leaders to change their style and policies in order to become more appealing for potential employees.

As previously mentioned, the management theory is also present in the health care system, especially in corporate medicine. An example of this management style is offering commission on top of a base salary, a widely used salary system in the veterinary health care system. The management theory is straightforward to understand and apply and can be very effective with the right type of employee. In the authors’ observation, less ambitious veterinary doctors, part-time employees, and employees who do not have major medical, managerial, or financial responsibilities (janitors, entry level receptionists, and entry level nurses) are more likely to perform well under transactional leaders. However, eager veterinarians who want to grow professionally, higher level administrative personnel, and, in general, more ambitious individuals are unlikely to express their full potential under this type of leadership.

The participative theory of leadership offers advantages to the veterinary health care system especially at small to medium sized organizations. Staff veterinarians of any experience and age will obviously feel much more motivated when directly involved in decision-making. The leader must be careful to gauge the allotted freedom according to the experience of the individual on staff. The risks are that of making young veterinarians feel overwhelmed and insecure, and that of making more experienced staff members feel condescended upon. A certain degree of participation is, based on the authors’ experience and observations, always appreciated, and received positively by all employees, veterinarians, nurses, administrative personnel, and barn crew.

Power leadership offers little advantage in the veterinary health care setting, as stated before, and therefore it will not be further discussed.

Working on establishing and maintaining connections, and using them at the right time, as advocated by the homonymous theory, has become increasingly important for veterinarians and veterinary leaders. Within a medium to large organization, it is important to choose the right people for the right tasks, and it is vital for the good outcome of the cases that communications between those people flow seamlessly. An example is that of a horse admitted after hours to a large equine referral practice for acute abdominal pain. The emergency service treats the horse conservatively based on the workup performed on admission. In the morning the horse’s clinical picture worsens and it gets transferred from the emergency service to the day surgery service for celiotomy. The anesthesia service also must be called to anesthetize the patient and assist until recovery is completed. Three services (emergency, surgery, and anesthesia) must communicate openly and efficiently for the sake of the horse’s life and wellbeing. This is much easier if the connections between those services are solid, friendly, established, and oiled, as it happens under an efficient connective leadership.

Transformational leadership is understood in terms of the leader’s influence on followers [49], and is currently a widely advocated leadership approach especially for the health care sector. The positive impact of transformational leadership on the quality of health care provided is substantial, as stated by Fischer [52], as it is associated with a better performance of the team and with improved patient care, although the mechanisms by which this happens are not entirely clear.

Transformational leaders are visionary, balanced, self-aware, and confident [53]. They generate commitment of the employees to the vision and ideal of the organization, encouraging them to exercise leadership and achieve exceptional results. The transformational leader discourages dependence by stimulating growth and development. The term “transformational” implies being visionary, thus, having ideas and aspirations on how things could be different and better, and then implementing these visions. As a consequence, transformational leadership is widely advocated especially for health care and social settings. It is to be noted, however, that transformational leaders display substantially more energy and enthusiasm than those around them, and therefore could be seen as pushy by their followers.

Research on transformational leadership [49,54] for health care setting suggests that this is the most useful theory for healthcare professionals, because transformational leaders are seen to encourage followers to go beyond goals related to self-interest and toward goals of the organization [49]. Transformational leaders build capacity by role modelling the core values of the organization, and they build a unifying purpose which the staff feels a part of. Where there is transformational leadership, staff are more engaged and more productive, and organizational goals are met more consistently [54]. In the authors’ opinion based on their experience and observations, this is also true in the veterinary health care sector.

Another aspect of what health care professionals face is crisis in health care, which poses a different set of leadership challenges compared to routine personal and public health care. Examples of crisis in public health care are the latest COVID-19 pandemic (2019 to present, still ongoing) or, in veterinary medicine, the Equine Herpesvirus epidemic (2021, 2022). The leadership skills required in these unique settings have been first examined by Deitchman [55] in their review and commentary.

A crisis is defined as a substantial threat to a social system and to the way it is structured, to its values, to its laws and regulations, threat that needs to be addressed quickly despite the uncertainty of times, through critical decisions [56]. A good leader in a time of crisis must withstand the enormous stress; must recognize that the crisis exists; must be able to make decisions, even unpopular ones, quickly and effectively and even in the absence of all necessary information; must be able to effectively and timely communicate; must be able to delegate without losing control.

Professionals employed in the public health sector are expected to lead responses to emergencies such as infectious diseases clusters, zoonotic diseases outbreaks, disease outbreaks in food animal impacting the food chain, or outbreaks of foodborne sicknesses. In this type of crises, however, the normal emergency measures are not effective in restoring the status quo [57]. Such crises can surge spontaneously or be initiated by hurricanes, tsunamis, draughts, extreme temperatures, and other natural causes; naturally occurring or laboratory-induced pandemics; biological terrorism; chemical spills; nuclear or radiological accidents. The authors have not been able to find in the literature any articles addressing the topic of health care leadership in critical circumstances and Deitchman was not either, according to their review’s citations. Deitchman, therefore, branched out and reviewed the crisis response in the following groups: flight captains, military leaders in extremis (meaning in a situation where the leader’s life and those of the team are perceived to be in danger), incident management team leaders such as firefighters or police officers, nuclear IMT commanders, and survivors of underground fires in mines. They observed that the surveyed disciplines display different requirements, challenges, training, working hours, work setting, and team size. A firefighting brigade on an accident scene may be composed by over 100 people, whereas a flight crew may be composed by as little as three. Firefighters and policemen are formally trained in the specifics of incident and crisis leadership. Military leaders and commanders of first response organizations receive formal training in incident leadership, but they may or may not have led a team in a real emergency situation before. Nonetheless it was noticed that effective crisis leaders seem to consistently display across the diverse settings some specific features, in that they have to be competent, decisive, self-confident, they need to be fast in receiving and processing external inputs, they have to communicate effectively, and they have to inspire trust in their team. Deitchman proposed therefore, based on the commonality displayed across the disciplines, a series of traits for the health care leader in times of crisis, which are: competence, decisiveness, situational awareness, acceptance of the inherited bidirectional communication responsibility towards both superiors and subordinates, coordination, inspiration of trust and self-confidence, and responsibility for the welfare of the team [55].

The crises the world experienced in the recent past in the global health system have challenged the existing training of health officers and leaders, and have demonstrated its inadequacy to withstand global emergencies such as the latest pandemic. In other words, it has become obvious that the traditional mix of basic management skills and medical knowledge is insufficient. Indeed, the large majority of the leadership has been taken by political forces and very little was left in the hands of health officers. As previously said, the public health environment benefits in normal times of a democratic, participative, and inclusive leadership which, however, may not be indicated for times of crisis, when the need arises of making decisions quickly despite the lack of complete information.

Because of this, some authors have advised for a more authoritarian decision-making process in terms of crisis, that should replace the conventional model in place in regular times [58]. Care should be taken, however, in promoting authoritarianism: these leaders in fact risk overlooking important contributions coming from the team, especially when team members have diverse, complementary areas of interest and expertise.

The truth is found probably somewhere in between the extremes of rigid mastery and never-ending search for general consensus. A leadership strategy that encourages open communication even in the face of crises seems to be more effective and safer [59]. The importance of open communication and approachability is exemplified by the experience of Captain Al Haynes in 1989. Captain Haynes and his crew managed to land a DC-10 despite the loss of its hydraulic controls. He gave credit to his team for using the so called CLR, crew leadership resource training. He admitted he would not have known better than any other crew member on that plane what to do, and therefore listening to the crew’s opinions was very important for a successful outcome [60].

Team performance in the health care sector during crises has not been thoroughly investigated and researched. However, it is logical to think that it would follow the same trend and be better when the leadership allows and encourages team contributions, as it is the case with transformational leadership. Health care leaders in times of crisis have to juggle between authoritarianism and democracy, as they wear multiple hats soliciting opinions, reviewing data and statistics, receiving expert opinions, and, finally, making and executing decisions even if not all pieces of the puzzle are present or coincide [55].

In summary, transformational leadership is about a joint vision, about prospective, about common values and direction, and inspiration for the future. Transformational leaders look beyond the daily tasks and goals and rather establish a direction and a path for the organization to accomplish its goals and objectives. Transformational leaders are people who not only do things right, they strive to do the right thing. The difference is that the focus of these leaders is on their vision and judgment rather than on mastering daily technical tasks. Therefore, these leaders are not focused on being efficient, but rather on being effective. Our impression is that transformational leadership is the only theory that is visionary and flexible enough to apply to the contemporary veterinary health care system and its never-ending set of challenges and changes, as it enhances the skills of every team member, it promotes their involvement and encourages them to go the extra mile. We feel that transformational leadership fosters the sense of belonging, it promotes necessary change following a broad-minded vision, and it prepares the individuals and teams for responsiveness in crisis situation, ultimately improving and enhancing the experience for employers, employees, clients, and patients (Figure 1).

In order to collect evidence to prove or to challenge our impressions, basic research is needed amongst human resources in the veterinary health care system, to investigate their needs, their fears, their goals, and their vison. This research has already begun in the small animal sector of the system [61], traditionally more advanced and more inclined to follow the progresses of the human medical world. The more traditional large animal sector is, on the other hand, still to be explored.

A first important step would be to administer human resources employed in large animal hospitals questionnaires modeled after the small animal works [61], to examine their conduct and retention time in relation to the type of practice they are employed in, and to investigate what variables influence them.

Motivated, competent, and professionally satisfied personnel should perform better and be retained longer, in turn ensuring better and more continuous patient care, as well as better outcomes for the patients and the sector they are part of. This speculation should be proven by prospective research that examines how different leadership strategies results in several objective and subjective parameters referred to all parties involved (personnel, patients, owners, industrial, and/or commercial partners).

Human resources are ultimately the most valuable asset of the veterinary business, and only an established leadership culture that considers every team member as valuable will withstand the challenges that the future of the large animal veterinary industry will bring.

**Figure 1 vetsci-09-00538-f001:**
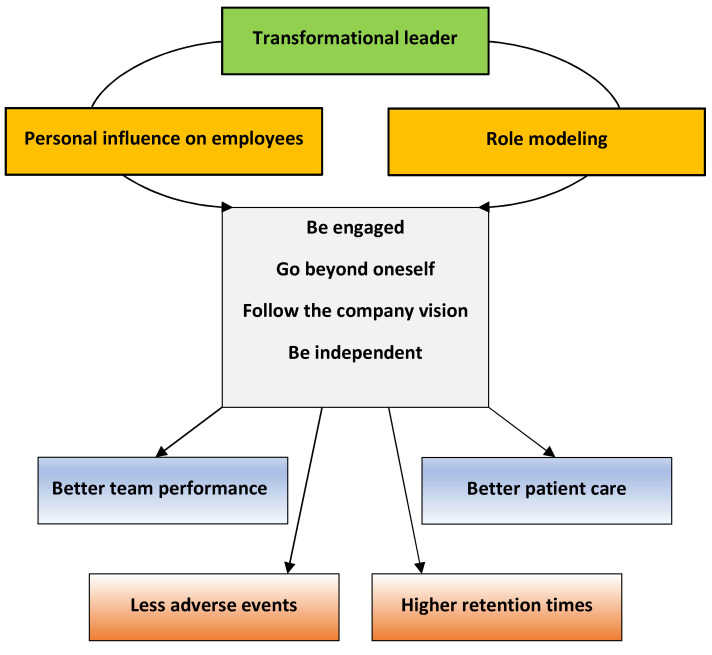
Potential effects of transformational leadership in health care.

## Figures and Tables

**Table 1 vetsci-09-00538-t001:** Leadership theories and their key points.

THEORY	KEY POINT
TRAIT THEORY	Leader’s innate characteristics
FUNCTIONAL THEORY	Leader’s functions
BEHAVIORAL THEORY	Leader’s styles
	Authoritarian
	Democratic
	Permissive
	Bureaucratic
CONTINGENCY THEORY	Leader’s ability to adapt to situations
MANAGEMENT THEORY	Rewards and punishments system
PARTICIPATIVE THEORY	Participations of the whole team to leadership
POWER THEORY	Power of the leader over his team
CONNECTIVE THEORY	Ability of the leader to connect people
TRANSFORMATIONAL THEORY	Leader’s vision

## Data Availability

Not applicable.

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
