# Peer review of "Leadership Theories and the Veterinary Health Care System"

_vetsci, 2022, doi:10.3390/vetsci9100538_

Round 1

Reviewer 1 Report

Dear authors,

I commend the authors for aiming to move veterinary practice forward by having a critical look at leadership theories and what the veterinary community can learn from these theories. However, it was unclear if this review is meant to advise veterinarians in terms of practice management (e.g., how to lead their employees), crisis management (e.g., how to lead a public health emergency team when there is a zoonotic health threat), how to interact with clients, or all of the above. I believe, making this very clear and structuring the article accordingly will be useful to the reader.

Further, I think the impact of this article can be increased by making some fundamental changes. Currently, the review of the different theories is rather ‘dry’ with no explanation of how they apply to the veterinary context (in contrast to what was ‘promised’ in the Abstract -line 24/25). Further, it remained unclear to me, which of these theories are obsolete, or at least irrelevant to veterinary medicine. The authors then conclude that transformational theory is most applicable to veterinary medicine. I believe there is a missed opportunity to provide the reader with more specific examples of what can be done by veterinary leaders to be transformational leaders, and this could build on Fig. 1.

Additional comments:

Based on the simple summary, it sounds like the focus is the human health care sector. I’d suggest making it more consistent with the Abstract

There are often very few references in the text to support the authors’ statements (e.g., Participative theory only has 1 reference in the entire section) 

I suggest being more specific about the applicability of a statement depending on the veterinarian’s focus of work. E.g., line 351: This sounds like this statement is limited to horses and/or pets, and the livestock sector is not considered.

Language could be more concise, especially in the discussion, where entire sentences could be deleted without losing any core information.

The Discussion appears fairly unstructured, e.g., the authors establish that transformational theory is most suitable for health care settings, then there is a page about crisis management, and then it comes back to transformational leadership.   

I would encourage the authors to refrain from using phrases that display their personal opinions such as “a very interesting aspect” (line 53) or ”in a very interesting review” (line 409)

Line-by-line comments:

Line 55/64/161 and elsewhere: Not all leaders are male…

Line 60/130 and elsewhere: Please aim to be gender inclusive (I would argue that is important for a good leader as well):  https://www.ccsu.edu/lgbt/files/preferredgenderpronouns.pdf I would suggest using “they”

Table 1: Is “Theory” missing for the last 2 entries?

Line 108: Unclear to me how, if they can be “developed” when they are not “taught” or “learned”. How else can they develop?

Line 345: Static in what sense? I would argue that major advances to the medical aspects of veterinary care have been made.

Line 348/349: A claim like this needs a reference, as I am sure many would disagree

Line 379 – what does ‘subjectively speaking’ mean?

Line 402/403: This is a big claim (that transformational leadership is the most useful theory for healthcare settings) and I would appreciate more information as to why this is the case.

Line 424=”authors”, Line 426=”author”, Line 430/437=”He” à Please be more consistent

Author Response

Dear Reviewer,

Thank you for your comments. We have largely edited the manuscript, and especially the discussion, to address them.

The gender has been addressed and the pronouns edited accordingly.

The word "Theory" has been added in Table 1.

The comment about previous line 108 has been addressed (see lines 115-118)

The comment about previous line 345 and 348-349 has been addressed in the edited discussion (see lines 363-378).

Line 379: removed.

Line 402-403: we have expanded the discussion to hopefully answer this concern as well. Also Figure 1 has been edited.

Reviewer 2 Report

Dear authors,

Mi comments below:

The study and application of leadership theories in animal sciences and veterinary health care are more important today than ever. Discussing advantages and bottlenecks and identifying the contexts of development is crucial. As set by several authors, the health care industry must improve attention while promoting fairness, transparency, and quality values (among others). 

Transformational leadership foster positive and inspirational changes in an organization; however, it comes with barriers and challenges depending on the (e.g., sociocultural context). Explaining how these theories may affect companies and societies is necessary to promote veterinary health care and science.

In my opinion, the manuscript entitled "LEADERSHIP THEORIES AND THE VETERINARY HEALTH CARE SYSTEM" points to the critical discussion on leadership in the veterinary health care system. In specific fields, veterinary remains a rural discipline that, due to cultural factors, has not changed as health care does. Another exciting aspect of this narrative paper is that it is straightforward to understand our future leaders' students and early carrier researchers.

The manuscript is generally well written. The table is concise and informative. The paper's originality, objectivity, currency, and coverage are average, but it may be amended. The authors seem to be a collaborative team working on veterinary practice and management, which offer both perspectives.

My general concern is that the leadership theories have already been defined and discussed. Beyond defining them, it would be more appreciated if you also write high-quality examples in the veterinary health care system with a list of two or three potential advantages and disadvantages/challenges/risks. This would give new novelty to the manuscript and make the message more captivating and understandable. Anything similar that you did in lines 266-268?

Moreover, in the discussion section, I suggest you focus more on a critical evaluation of the transformational theory and how this may lead to further investigation and approaches toward veterinary health care. I am not inviting you to change everything; I would like you critically discuss the theory in the proper context.

I would recommend updating the paper and looking for recent literature. There is exciting literature in recent years about transformational leadership (e.g., remote healthcare leadership, patient safety outcomes, work engagement). Probably this will change your perspective of the sector or not. After that, please ensure that the format of the references follows the journal's recommendations (https://www.mdpi.com/authors/references).

Other concerns in detail are as follows: 

1) Please include a short final chapter about perspectives, future research in the field, and how this would impact the veterinary health care system, animal welfare, environment, and society.

2). Please improve figure 1. Instead of the only characteristics, a figure summarizing the whole discussion would give the reader a better overview of the transformational leadership in veterinarian health care.

3). Please cross-check the list of references revising for the inclusion of predatory journals in any of your references.

4) Between lines 52-53, please write a paragraph justifying why leadership theories should be specially discussed in veterinarian health care. Why are they necessary in the sector?

5) In the paragraph comprising lines 327-340, please also provide the disadvantages of the transformational theory. And then the example I mentioned before.

6) The text between lines 357 and 387 may be shortened. 

7) Please improve the quality of your figure 1 substantially. 

8) Please focus on the figure flowing. Use arrows to delineate the sources and the targets. What triggers what? Potential outcomes?, Perspectives? More color would also be appreciated.

Best regards

Author Response

Dear Reviewer,

Thank you for your comments. We have largely edited the manuscript, in particular the discussion, to address them.

A final chapter on perspectives and future research has been added (609-613). Figure 1 has been replaced by a figure that illustrates the positive cascade effects of good transformational leadership in the veterinary health care.

About point 4, this part of the manuscript has been edited to accommodate comments from all reviewers.

Disadvantages of transformational leadership have been listed.